# Perception Toward Wolves Are Driven by Economic Status and Religion Across Their Distribution Range

**DOI:** 10.3390/ani15091196

**Published:** 2025-04-23

**Authors:** Dipanjan Naha, Stefanie Döringer, Marco Heurich

**Affiliations:** 1Independent Researcher, Kolkata 700047, India; dnaha234@gmail.com; 2Department of National Park Monitoring and Animal Management, Bavarian Forest National Park, FreyungerStraße, 94481 Grafenau, Germany; stefanie.doeringer@npv-bw.bayern.de; 3Department of Geography and Regional Research, University of Vienna, Universitätsstraße, 1010 Vienna, Austria; 4Faculty of Environment and Natural Resources, University of Freiburg, 79085 Freiburg im Breisgau, Germany; 5Department of Forestry and Wildlife Management, Inland Norway University of Applied Sciences, Evenstads Vei 80, 2480 Koppang, Norway

**Keywords:** attitude, carnivore, coexistence, social drivers, review, tolerance

## Abstract

Understanding people’s perception of large carnivores is important to address issues with human–carnivore coexistence. Despite considerable research and efforts being undertaken to study and conserve grey wolves, research on the global perception toward the species is mostly site-specific. Realizing the gap in our understanding of the perception toward grey wolves across their distribution range, we conducted a systematic review to identify the dominant perception and major factors influencing them. Our study shows that the dominant perception toward wolves is negative, with the predominant religion and economic status of the country being the significant factors. This information helps identify regions where conservation efforts should be concentrated to improve overall tolerance toward the species. A global strategy to conserve the species should consider the diverse social factors and differences in human perceptions and attitudes across geographic regions.

## 1. Introduction

The global decline of large carnivore populations is primarily driven by the loss of habitats, depletion of prey, and direct persecution by humans [1]. Large carnivores are integral to megafaunal communities and shape ecological dynamics through trophic cascades and maintaining ecological balance [2]. They can regulate herbivore and mesocarnivore populations, influence prey behavior and activity, and ultimately help sustain biodiversity [1]. The dietary requirements, movement behavior, and resource use of carnivores often result in conflict with humans [3]. Such interactions lead to negative impacts on people’s safety and livelihoods [4]. The complex interrelationships between ecological and social attributes determine the outcome of human–carnivore interactions. Cultural intolerance, economic losses through livestock predation, and perceived risk increase the fear and negative attitudes toward large carnivores. These factors lead to retaliatory killing of large carnivores, increasing the threat of local extinction and the loss of the ecosystem services they provide [5]. Different human communities perceive carnivores differently, impacting how carnivores share space with humans. Consequently, the future persistence of large carnivores depends on understanding the sociological dimension of human–carnivore interactions and developing effective mitigation measures.

Historical narratives and human–nature relations play an important role in shaping perception toward large carnivores [6]. Large carnivores were eradicated from areas where they were historically perceived as threats to human safety and livelihoods. However, they were tolerated in areas where humans perceived them positively and associated them with spiritual values. Lions (*Panthera leo*), cheetahs (*Acinonyx jubatus*), and African wild dogs (*Lycon pictus*) have been historically persecuted across Africa due to the perceived threat to livestock, whereas in South Asia, tigers (*Panthera tigris*) and leopards (*Panthera pardus*) have coexisted with humans due to a strong cultural reverence toward wildlife. The diversity of attitudes toward carnivore species and their variation across regions highlights the importance of considering local beliefs and perceptions when developing conservation strategies and understanding human–wildlife interactions. Early human societies regarded wolves with respect and admiration. However, during the transition phase from hunting to agriculture-based societies, wolves were persecuted across Europe and North America by farmers who viewed them as a threat to their safety and economic prosperity [7]. Wolves symbolized vast wilderness areas occupied by game, and the killing of wolves showcased dominance over nature, resulting in the conversion of those areas to farmlands and the displacement of indigenous communities [8]. However, with the development of wildlife conservation laws and policies in the 1970s, wolf populations started recovering in parts of Europe and North America [9,10]. Compared to Europe and North America, information on wolf distribution, status, and human–wolf interactions are relatively lacking in East and Central Asia [11,12].

Globally, major religions such as Christianity, Islam, Hinduism, Buddhism, and Judaism have had an influence on human societies and their relationship with nature [13]. The spread of Christianity and Islam and the associated cultural, economic, and religious factors influenced human–wolf relations and the survival of wolf populations [7]. Christianity presented an anthropocentric version of the world where humans were considered superior to all other life forms and wolves were regarded as threats to livelihoods and spiritual purity. Unlike Christianity, Islam does not prescribe human dominance over nature or exploitation of natural resources. Religion and cultural factors have been globally documented to impact people’s values and beliefs, ultimately shaping perception toward large carnivores [13,14,15,16]. Compared to Europe and North America, several Asian countries exhibit a cultural reverence toward wildlife. Tigers, leopards, and snow leopards (*Panthera uncia*) are often associated with powerful deities, indicating a positive perception toward large carnivores [17,18,19]. Thus, people from Asia generally have positive cultural beliefs regarding wildlife and are more tolerant of predators compared to Western societies.

The grey wolf (*Canis lupus*), hereafter wolves, is one of the most widely distributed large carnivores and inhabits diverse ecosystems across North America, Europe, and Asia [20]. Wolves are apex predators and play an important role in regulating prey populations and maintaining the functioning of natural ecosystems [10]. The relationship between wolves and humans played a significant role in the development of human societies across the Northern Hemisphere. Wolves have diverse symbolic meanings, reflecting their complex cultural representations across human societies. The cultural perspective toward wolves ranges from reverence, symbolism, admiration, and respect to fear and hatred [7]. During prehistoric times, indigenous communities admired wolves due to their hunting strategies and social structure and associated them with their own survival tactics. The advent of agriculture, ownership of land, and the spread of major religions such as Christianity altered wilderness areas across Eurasia and North America. Wilderness areas were transformed into cultural landscapes, and wolves were considered major threats to agrarian livelihoods [21,22]. Wolves were often depicted with supernatural powers in relation to the devil and witchcraft in popular folklore and literature in Europe. Certain wildlife species generate disproportionately extreme emotions compared to their actual threat, highlighting the complexities of attitudes and their impact on human societies [23]. The negative cultural perception of wolves affected human–wolf relationships and their cooccurrence with humans. The perceived risk from wolves was significantly higher than the actual risk they posed to humans [23]. In rural areas, the predation of livestock and the economic damage had severe implications for livelihoods, fostering public intolerance toward wolves [24]. Due to this negative image, wolves were historically persecuted across North America, Europe, and parts of Asia [10,20]. Thus, human-caused mortality is the primary factor behind the global decline in wolf populations [25]. Although wolves were eradicated from large areas of their historical range, they are currently recolonizing parts of their former range, with an overall increase in regional and global populations. This recolonization process is supported by legal protection, conservation programs, management practices, and law enforcement measures aimed at reducing the costs of coexistence with wolves. These measures are also interrelated with increasing public tolerance and a more positive perception of the species [26,27]. Media plays an important role in how people base their opinions around social issues, and the coverage of topics around large carnivores plays a crucial role in how they are portrayed in human societies. In areas experiencing recent colonization or reintroduction of wolves, media coverage of topics around human–wolf conflicts could trigger negative public perception toward the species. There has been considerable research on the role of the media in shaping people’s opinions toward wolves [28,29,30].

Contemporary studies often report that human–wolf conflict is reflective of the urban–rural divide and highlight the dominance of urban residents over rural ones, thereby portraying the marginalization of rural actors and rural interests [31,32]. Additional research further highlights this cultural divide and difference in perspectives regarding human–wolf cooccurrence between urban and rural residents [33]. Rural communities often become upset by damage caused by wildlife, which they perceive as being protected or imposed by powerful urban elites and policymakers [32,34]. Hence, policymakers need to address this underlying social tension, including understanding the difference in acceptance of predators between the two groups [31]. Their impacts on livestock also shape the difference in tolerance toward wolves. Livestock constitutes an increasingly important prey base for large carnivores within cultural landscapes. Considering that livestock rearing is a major livelihood strategy across rural areas, depredation by wolves represents a significant occupational threat to income generation by farmers [35]. Consequently, financial losses due to wolf attacks remain a major driver of perception toward wolves [36].

Both perception and attitude determine the extent to which humans are willing to coexist with wildlife. Based on [37], perception is “about receiving, selecting, acquiring, transforming and organizing the information supplied by our senses”. Perceptions are not just based on personal experiences, but social and cultural norms and beliefs can influence them [6]. Historical events and cultural beliefs can influence our perception toward a certain species. Attitude is defined as “a tendency expressed by evaluating a particular entity with some degree of favor or disfavor” [38]. Thus, perception may impact how people understand/interpret the world and make decisions [39]. Attitude is also defined as “dispositions or tendencies to respond with some degree of favorableness, or not, to a psychological object, the psychological object being any discernable aspect of an individual’s world, including an object, a person, an issue, or behavior” [40,41]. Attitudes can be a combination of thoughts, feelings, or opinions about a particular object and can be regarded as positive or negative thoughts, feelings, or behavior toward the object [39]. Although attitude and perception are broadly similar, attitudes include the evaluation of a subject or topic based on perception. Both these concepts together determine behavior, intentions, and actions toward wildlife [39]. Although most studies highlight the extent of damage, attitudes and perceptions are often the strongest predictors of behavioral intentions toward carnivores [6]. Attitudes and perceptions are based on several factors, such as past experiences, values, norms, beliefs, knowledge, motivation, and socioeconomic characteristics [42]. Consequently, an understanding of the interrelationships between attitudes and perceptions and the complex web of socioeconomic factors, cultural beliefs, symbolism, and financial costs of sharing space with wildlife form the basis for developing appropriate conservation measures [6].

The recolonization process and the resurgence of wolves within shared regions provide an ideal opportunity to examine the heterogeneity in societal acceptance of a widely distributed large carnivore and understand how their perception varies across cultural groups, communities, geographic regions, and continents. To the best of our knowledge, no systematic quantitative review has been conducted to understand the major sociocultural drivers and heterogeneity in the perception of wolves across their current distribution range. Designing a global conservation strategy without documenting the variation in perceptions toward wolves is challenging. We aimed to compile and evaluate the current available information on their public perception and possible linkages with global conservation efforts. We examined all research articles published between 1980 and 2023 to assess the overall perception of wolves and identify diverse sociocultural factors shaping their perception. Finally, we identified research gaps and provided recommendations to improve human–wolf coexistence across their distribution range.

We answered three basic research questions through this study: What is the global perception of grey wolves, and how does it vary across continents, cultural, and economic groups? What are the major factors driving perception toward wolves? What are the potential gaps in human dimension research toward wolves? Using data generated from the systematic review of published research, we tested the following three hypotheses: negative perceptions are more widespread in (1) rural compared to urban communities, (2) regions dominated by Christianity and Islam as major religions, and (3) groups experiencing financial damage due to wolf attacks on livestock.

## 2. Materials and Methods

We systematically reviewed peer-reviewed English-language articles on attitudes and perceptions toward wolves. Wolf refers to gray/grey wolf (*Canis lupus*), following the guidelines of [43]. The major aim of our review was to summarize and appraise results from the extensive body of research. Articles were also identified from the reference list of each publication. The scientific literature was identified using two electronic databases (Web of Science and Google Scholar) and a reference-mined (bibliographies/works cited/reference list of relevant articles) approach. Our major interest was to document the variation in perceptions after the recolonization and resurgence of wolves across new areas in North America and Eurasia, and hence the search was restricted to articles between January 1980 and October 2023. The word combination used for the search included either “gray/grey wolf” or “Canis lupus” with any of the following keywords or phrases: belief, attitude, perception, tolerance, culture, values, social drivers, acceptance, and emotions. The literature search was conducted by adding, one by one, the names of all range countries (within the IUCN grey wolf distribution range) and the three continents (North America, Europe, and Asia), in combination with the keywords mentioned above, to obtain articles for the entire global wolf distribution range. The initial search and 1st screening returned a total of 301 articles (Figure 1). These articles were then assessed manually for content and the methodology used in the study. After the 1st screening, we read all articles and independently reviewed them to remove duplicates and ensure consistency based on the research topic (all authors participated in this). The 2nd screening resulted in a total of 124 articles. We included studies based on the criteria that the authors used a quantitative approach and excluded reviews, qualitative research, and conceptual papers. Our focus was on studies that quantified attitudes toward wolves, and hence we removed studies that examined attitudes using qualitative methods. We included comparative studies conducted in different countries, analyzed the results separately, and excluded grey literature to avoid publication and methodological bias. Grey literature refers to dissertations, theses, and project reports and was excluded because they do not undergo a formal review process. Book chapters and conference papers were also excluded from the review for the same reasons. The final screening resulted in a total of 118 articles.

Finally, the studies that quantified attitudes and perceptions were considered for further data extraction and analysis. After the final screening, we extracted data from 118 research articles conducted in 35 countries between 1980 and 2023 (Figure 1, See Appendix A for the PRISMA reporting checklist and Appendix A for the final dataset). We followed the PRISMA guidelines for screening and data extraction. The articles considered for data extraction included surveys using different quantitative methods, such as closed and open-ended questionnaires as well as email-based and postal surveys.

The wording of the questions related to attitudes and perceptions toward wolves varied between studies. Studies used different questions to investigate attitudes and perceptions. We considered this diverse range of questions, broadly related to the major theme of perceptions toward wolves. Some studies used even and uneven Likert scales, whereas others used ranks, categories, or scores to rate the attitudes/perceptions of respondents toward wolves. For studies that used several questions in their surveys, we considered the question regarding attitudes/perceptions toward wolves as the primary response. We categorized perceptions into three major groups (positive/negative/neutral). We only considered the dominant perception reported in the study. A perception was dominant only if ≥50% of the respondents reported the same perception in the study. For studies reporting multiple perceptions, we considered the most dominant perception for further data extraction and analysis.

For each article that met our criteria, we extracted a set of primary variables (Table 1) and created a database. Information on the age, gender, and education of respondents was missing for several studies, and hence we did not consider them further in the analyses. Data included the spatial and temporal coverage of the study site between 1980 and 2023, and spatial data were processed in ArcGIS 10.5. No studies between 1980 and 1984 matched our search criteria, and hence the period for publications was categorized into two decades: Decade 1 (1985–2005) and Decade 2 (2006–2023).

We conducted a chi-square goodness-of-fit test to check for significant differences in the number of publications between decades. This was performed to check for progress in research based on the overall temporal trend of publications on this topic across decades.

We recorded information on the methods and properties of the included surveys (i.e., survey design, sample sizes, and sampling methods). The name of the first author was also recorded for each article. We recorded the year and country in which the survey was conducted and the name of the journal in which the study was published. After reading through all articles, we categorized studies into countries, continents, and cultural regions (Eastern Chinese, Indian, Slavic Russian, Western European, and Islamic). The cultural regions and the countries were defined as (i) Eastern Chinese—China, Mongolia, Japan, Nepal, Korea, and Myanmar; (ii) Indian—India, Bhutan, Bangladesh, and parts of SE Asia; Slavic Russian—Russia and several countries included in the former Soviet Union; Western European—all European and North American countries; and Islamic—Pakistan, Iran, Central Asia, the Middle East, and parts of North Africa. We compiled the number of studies and estimated the overall proportion for the three different categories (positive, neutral, and negative) of perceptions toward wolves. We conducted a chi-square goodness-of-fit test to check for significant differences in perceptions between the three categories.

We assigned each country an economic status based on the World Bank classification. We categorized each study into yes and no categories based on the presence or absence of livestock losses (economic damage) from wolves. We created ggplots to visually represent the relationship between perceptions toward wolves and continents, as well as the income status of countries. We combined all studies and considered a set of 6 categorical predictor variables (community, locality, religion, damage to wolf attacks, cultural group, and economic status) and used an ordinal logistic regression model (Perception~Community + Locality + Religion + Damage due to wolf attacks + Cultural group + Economic status) to analyze the community/people/group and their perceptions toward wolves. We used the MASS package in R for the ordinal regression. Statistical significance was determined at *p* < 0.05. Locality was categorized into 3 groups: rural, urban, and a mixture of rural and urban. A mixture of rural and urban areas included small towns interspersed within rural areas. Religion was categorized into 5 groups: Buddhism, Hindu, Christianity, Muslim, and Shinto. Continents were categorized into 3 groups, Asia, Europe, and North America. Studies that reported wolf attacks on livestock were labeled as “1”, whereas those that did not experience any attacks were labeled as “0”. The stakeholder groups were categorized as follows: hunters, livestock farmers, livestock herders, pastoralists, students, schoolteachers, and local residents. The measures of association were checked using likelihood ratio tests. We checked for multicollinearity among dependent variables and used the variance inflation factors (VIFs), where VIFs less than two implied the absence of collinearity. The pseudo-R^2^ statistics were estimated using the package DescTools in R. All analyses were conducted in R 4.2.0.

## 3. Results

### 3.1. Perception Toward Wolves

Fifty-four percent of the studies reported negative perceptions, 38% reported positive perceptions, and only 8% reported a neutral perception toward wolves (χ^2^ = 32.72, *p* < 0.05, df = 2).

### 3.2. Summary of the Spatial and Temporal Scale of Studies

Overall, the majority of publications on attitudes/perceptions toward wolves are from Europe (43%), Asia (34%), and North America (23%) (χ^2^ = 6.02, *p* < 0.05, df = 2, Figure 2). In Asia, the majority of studies were conducted in Pakistan (15%), India (5%), and Iran (4%); in Europe, the majority of studies were conducted in Norway (12%), Sweden (8%), Germany (6%), and Italy (4%); and in North America, majority of studies were conducted in the US (23%) and Canada (2%).

The number of publications exhibited an increasing trend from 2013 onward, with differences between years (Figure 3). The first twenty years, between 1985 and 2005, resulted in a total of 15 publications comprising only 13% of all research articles. The highest number of publications in period 1 was recorded in 2005 (N = 4). However, the next twenty years, between 2006 and 2023, resulted in a total of 103 publications comprising 87% of all research articles (χ^2^ = 59.29, *p* < 0.05, df = 1). The highest number of publications in period 2 was recorded in 2020 and 2021 (N = 16).

The highest number of publications (N = 49) were from Europe, comprising 41% of all research articles, followed by Asia (N = 40), with 34%, and North America (N = 29), with 25% (Figure 4).

### 3.3. Studies Across Continents

In Asia, the dominant perception toward wolves was negative, whereas it was positive in Europe and North America (Appendix A). Eighty-five percent of the studies from Asia and forty-one percent from North America reported negative perceptions of wolves. Fifty-six percent of the studies from Europe and forty-eight percent of the studies from North America reported positive perceptions toward wolves. The age of respondents in the studies ranged between 9 and 80 years.

### 3.4. Income-Wise Perception Toward Wolves

The perception of wolves was negative in countries with low, lower, and lower-middle income levels and positive in countries with high and upper-middle income levels (Appendix A). Eighty-five percent of the studies from regions in lower-middle-income countries and one hundred percent of the studies from regions within low-income countries reported negative perceptions toward wolves. Fifty-two percent of the studies from regions in high-income countries and twenty percent of the studies from regions in upper-middle-income countries reported positive perceptions toward wolves.

### 3.5. Perception Based on the Locality

Sixty percent of the studies reported that urban residents had positive perceptions, whereas sixty-nine percent of the studies reported that rural residents had negative perceptions toward wolves. Eighty-eight percent of studies reported that livestock farmers had negative perceptions, one hundred percent reported that livestock herders had negative perceptions, and sixty-seven percent reported that agropastoralists had negative perceptions toward wolves. Sixty percent of studies reported that hunters had positive perceptions, one hundred percent reported that schoolteachers had positive perceptions, and fifty-seven percent reported that students had positive perceptions toward wolves.

### 3.6. Perception Across Cultural Regions

Perceptions toward wolves were negative in Eastern Chinese, Islamic, and Indian cultural regions, whereas they were positive in the Western European cultural region. Eighty-seven percent of the studies from the Eastern Chinese cultural region, fifty percent from the Indian region, and eighty-nine percent from the Islamic region reported negative perceptions toward wolves. Fifty-four percent of studies from the Western European region and forty percent from the Slavic Russian cultural region reported positive perceptions toward wolves.

### 3.7. Perception Based on Major Religious Groups

All (100%) of the studies from countries with Hinduism as the predominant religion, 89% of studies from countries with Islam as the predominant religion, 50% of studies from countries with Shinto as the predominant religion, and 38% of studies from countries with Christianity as the predominant religion reported negative perceptions toward wolves. A total of 53% of the studies from countries with Christianity as the predominant religion, 25% from countries with Buddhism as the predominant religion, and 7% from countries with Islam as the predominant religion reported positive perceptions of wolves. A total of 50% of the studies from countries with Shinto as the predominant religion, 12.5% from countries with Buddhism as the predominant religion, and 9% with Christianity as the predominant religion reported neutral perceptions toward wolves.

### 3.8. Model Explaining Major Predictors of Perception

The ordinal logistic regression model revealed that the predominant religion of the region and the economic status of the country were significantly associated with perceptions toward wolves (Table 2). We hypothesized that people/groups living in rural areas would have a negative perception toward wolves. This hypothesis was rejected, as our results suggest that locality was not a significant predictor of perception toward wolves.

We hypothesized that people/groups living in regions with Christianity and Islam as the predominant religion would negatively perceive wolves. Our hypothesis was rejected, as the perception toward wolves was negative in countries with Hinduism, as the predominant religion. The stakeholder group with Hinduism as their predominant religion, when compared to groups with Islam and Christianity as their predominant religions, were associated with a higher likelihood of having a negative perception toward wolves. We hypothesized that people/groups experiencing financial damage due to wolf attacks on livestock would have a negative perception. This hypothesis was rejected, as our results suggest that financial damage was not a strong predictor of perception toward wolves.

People/respondents in low-income countries had a higher likelihood of having a negative perception toward wolves compared to people in lower-middle-income and upper-middle-income countries. The Nagelkerke and McFadden’s pseudo-R^2^ values for this model were estimated at 0.53 and 0.33, respectively.

## 4. Discussion

We synthesized information from 118 studies to examine the global perception toward wolves. The findings suggest that the predominant religion of a region and the economic status of the country where the study was conducted are the major predictors of perception toward the species. We did not observe any significant difference in perception toward wolves between rural and urban respondents, which was not consistent with the first hypothesis. We found that the perception toward wolves was negative in countries with Hinduism as the predominant religion, which was in opposition to hypothesis 2. Hypothesis 3 was not supported, as we did not document the damage due to wolf attacks as a significant predictor of perception. The variation in perception between respondents from different economic backgrounds could be related to the socioeconomic disparities between countries and their impact on conservation-oriented attitudes toward wolves. Our findings show that the perception of wolves across European and North American countries was a mix of positive and negative, which is different from the traditional narratives. A few studies involved research in Eastern European and Central Asian countries, with the majority of the studies being conducted in parts of Europe, North America, and South Asia.

### Major Drivers of Perception Toward Wolves

We identified the predominant religion and economic status of the country where the study was conducted as major drivers of perception toward wolves. The respondents interviewed in the study could differ from the overall population of the region and country in terms of economic status and religious affiliation. Despite this, economic prosperity is closely related to overall well-being [44] and plays an important role in how human society or communities perceive wildlife. Religion has often been an underexplored aspect of conservation ecology [45], and our results show that people from areas where Hinduism is the predominant religion tend to have more negative perceptions toward wolves. Generally, Hinduism is associated with reverence and respect toward nature, the environment, and wildlife. Unlike other rare mammals, the negative perception of wolves in regions dominated by Hinduism could be due to their higher perceived risk, lesser spiritual and cultural significance, occurrence within multiuse areas, lack of conservation efforts, economic benefits for communities, and the historic nature of human–wolf conflicts, including attacks on humans [46,47]. Wolves lack a strong positive cultural representation in South Asian folklore, and this narrative intensifies the negative perception toward the species [48]. Wolves have a complex and multifaceted symbolic role in Hindu mythology and culture, depicting both fear and respect toward the species. In the Annapurna Conservation Area, Nepal, ref. [49] reported that most people had a negative perception of wolves. In the Upper Spiti region, India, ref. [50] reported that people generally had a negative perception of wolves. Similarly, in Rajasthan, western India, ref. [51] reported that respondents had a predominately negative perception toward wolves. These studies suggest that, unlike other large mammals, there seems to be no positive cultural reverence toward wolves in countries dominated by Hinduism as the predominant religion. Thus, the predominant religion of the country should be considered an important feature for understanding the underlying sociopolitical processes shaping human–wolf relations and developing policies related to wolf conservation.

We did not find any significant difference in perceptions between urban and rural residents. Several studies have reported that urban residents generally tolerate wildlife [52]. On the contrary, the rural respondents/groups perceive wolves as a threat to their livestock and livelihoods and the restrictions they pose on hunting on communal and government lands. However, studies have also documented that rural people are positive about the conservation of wolves, highlighting the complexities of human–wolf relations. A study in 2016 [53] documented that there was an attitudinal divide between urban and rural residents toward wolves in Sweden. Rural people tended to have more negative attitudes, and this perceived difference was explained by a combination of factors, such as political alienation, proximity and direct experience with wolves, and underlying sociopolitical tensions between the two groups. In Utah, USA, ref. [54] reported that both urban and rural residents had a positive perception toward wolves. Similarly, in the Dauria ecoregion, Russia, ref. [55] reported that rural communities had both neutral and negative perceptions toward wolves. Wolves could symbolize a lack of social identity, restrictions on economic prosperity, and the imposition of environmental protection measures by the urban community that do not involve the rural community in decision-making and management. We understand that there could be site-specific perception differences between urban and rural groups. However, these differences are not significant on a global scale; hence, conservation programs should consider the well-being of both groups and focus on resolving the underlying social tensions. These results are in accordance with previous studies, which have reported that conservation conflicts are often site-specific and are often difficult to generalize across species, conservation issues, and large geographic regions [5]. Long-term coexistence programs should address the needs and concerns of both communities and actively engage them in participatory processes, such as planning and decision-making, for the conservation of wolves.

While some studies have reported that negative experience (livestock depredation) reduces tolerance and increases negative perceptions [56], we did not find the presence or absence of livestock losses to be a significant driver of perceptions toward wolves. This finding is in accordance with some of the published literature that suggests that economic damage by wildlife is not a significant predictor of attitudes toward a species [6,57]. Although the perception of economic damage varies between occupation groups [58] or in relation to the exposure time to carnivores, our results suggest that stakeholder type was not a significant predictor of perceptions toward wolves globally. Despite the lack of statistical significance, livestock farmers, herders, and agropastoralists had negative perceptions, whereas hunters, teachers, and students reported positive perceptions toward wolves. In the Nepalese Himalayas, [59] reported that agropastoralists had a predominantly negative perception toward wolves. A study conducted in Latvia [60] reported that the public and hunters had both positive and neutral perceptions toward wolves. Our finding suggests that broadly, negative attitudes/perceptions toward carnivores are not entirely limited to economic damage or specific stakeholder groups, and measures aimed at improving tolerance should target the well-being of all groups while implementing coexistence measures. Previous research suggests that the mitigation of economic damage to wildlife might not have a long-term impact on conflict resolution [6], and hence measures aimed at mitigating damage to wolf attacks might not improve the attitudes of local communities. We recommend a more holistic approach, where human–wolf coexistence programs use a multifaceted interdisciplinary approach addressing the complex problems around human–wolf cooccurrence and consider the interaction between socioeconomic, ecological, and cultural factors to develop appropriate strategies. We understand that there could be a bias toward studies measuring the negative economic impacts of wolves [61] and suggest a stronger research focus on quantifying positive socioeconomic benefits and intangible costs of living alongside large carnivores.

The global distribution of grey wolves varies across regions, with significant populations in North America, Europe, and Asia. Most of the wolf populations within Europe are distributed in Russia and countries included in the former Soviet Union (Romania, Ukraine, Belarus, Bulgaria, and Georgia). Most of the grey wolf population in North America occurs in Canada and parts of the USA. Most of the wolf population in Asia occurs in India, Iran, China, and parts of Central Asia [62]. Research on the human dimensions aspect of wolves shows significant gaps in studies across their distribution range. The country with the highest number of studies was the United States, followed by Pakistan, Norway, Germany, and India. The spatial difference in the quantity of human dimension research conducted on this topic could be due to the availability of research funding within specific countries, greater participation of the public on wolf-related issues, or the presence of long-term projects on human–wolf interactions in these regions [63]. We suggest that these geographical biases be reduced by conducting studies in European and Asian countries where wolves occur with humans. Russia and countries from the former Soviet Union host a substantial wolf population; however, there is a large deficiency in human dimension research from this region, and such areas should be prioritized for future human–wolf coexistence programs. This trend could also be due to the overall lack of wolf-related literature in English from this region.

The temporal trend in the progress of research on this topic suggests an increase from 2013 onward, with the highest number of publications in 2020 and 2021. The highest number of publications in 2020 and 2021, comprising 29% of all articles, could be related to the global lockdowns during the COVID-19 pandemic and the time available to researchers to write scientific articles [64].

The cultural symbolism of carnivores varies across geographic regions and plays an important role in shaping tolerance toward wolves. In ancient European and North American societies, wolves held positive spiritual symbolism related to power and intelligence and were treated with respect and admiration. Wolves were also associated with strength and honor, highlighting their importance in the cultural practices and beliefs of tribal communities [7]. With the spread of Christianity and Islam, wolves were widely disliked across geographic regions worldwide. As human societies progressed from animistic beliefs to mainstream global religions, the perception of wolves changed from a positive narrative to a negative one. The dominant negative perception of wolves in major global cultures highlights their significance in shaping attitudes, emotions, and human behavior, thereby impacting conservation efforts and the contemporary nature of human–wolf interactions. Our results suggest no significant differences in perceptions toward wolves between diverse cultural regions. These findings highlight the dynamic nature of cultures and how they have changed over time, thereby impacting people’s relationship with wolves and their perceptions toward the species. This further provides an opportunity for the research community to examine how cultural perspectives toward wolves have developed or changed over time across different geographic regions.

## 5. Conclusions

Despite the widespread distribution of wolves and the challenges around human–wolf co-occurrence within shared landscapes, there is very limited information on the global perception toward the species. The long-term survival of wolves and several other large carnivores will depend on understanding the factors affecting people’s perceptions and improving overall societal tolerance across their distribution range. The results of this review suggest heterogeneity in how people perceive wolves across the globe. Our study identified the major social drivers affecting perceptions toward the species. We demonstrate the role of the predominant religion and the economic status of the country as major determinants of perceptions toward wolves. Human–wolf coexistence strategies might differ between geographic regions, depending on the needs and priorities of the local communities, government, stakeholders, wildlife agencies, and the nature of human–wolf relations.

This review highlights the need to examine the factors determining differences in perceptions between economic and religious groups, diverse stakeholders, and cultural regions while effectively designing strategies for ensuring human–wolf coexistence globally. The lack of adequate research on this topic can lead to incomplete insights into the complexities of human–wolf relations, thereby hindering the development of effective coexistence strategies. Interdisciplinary studies integrating environmental, social, economic, and cultural aspects should be conducted to address the knowledge gaps and develop appropriate human–wolf coexistence strategies. Our results suggest that conservation efforts and human dimension research must focus on the Far East and other Asian countries where wolves cooccur with humans.

Contrary to historical narratives, we found that perceptions toward wolves were negative in Asian countries. This was unexpected, considering the cultural reverence toward carnivores and a long history of human–wolf co-occurrence within Asian landscapes. Such findings also highlight the need to improve conservation efforts and cultural stigma toward the species through targeted interventions. The findings suggest that governments, community leaders, conservationists, and other stakeholders should collaborate in ensuring human–wolf coexistence within the region. Given that how people perceive carnivores varies, efforts must be made to collect baseline data from countries where human dimension research on wolves is limited. More longitudinal and comparative studies should be initiated to understand the differences and similarities in perceptions across localities. In-depth studies should be conducted to understand the potential role of predominant religious, cultural, and economic factors on perceptions and how they impact human–wolf interactions across their distribution range. We understand the limitation of studies using quantitative methods to document perceptions toward wolves. To overcome the methodological bias associated with contemporary studies, researchers must use a combination of quantitative and qualitative methods to obtain a holistic view of human–wolf relations. The top priorities for future human–wolf conflict mitigation programs should focus on developing regions of Asia, and cross-cultural collaborations could be initiated between Europe, North America, and Asian countries to develop effective management strategies.

Human–wildlife conflicts are a major global driver of biodiversity loss, especially in areas where wildlife co-occurs with humans. Such negative interactions are always underpinned by social conflicts between different groups of people, impacting attitudes toward conservation and tolerance toward wildlife. This is especially challenging when species range beyond protected areas into multiuse areas. The recolonization and occurrence of wolves within human-dominated landscapes pose potential threats to income generation by farmers and livestock-based livelihoods [35]. As wolf populations increase and expand into new areas, conflict with humans remains a major challenge for human–wolf coexistence. To ensure the persistence of wolves within shared landscapes globally, efforts must be made to understand the complexities of human–wolf interactions and address the underlying social issues affecting the well-being of diverse local communities. This could be achieved through increased public participation in decision-making for wolf conservation programs.

## Figures and Tables

**Figure 1 animals-15-01196-f001:**
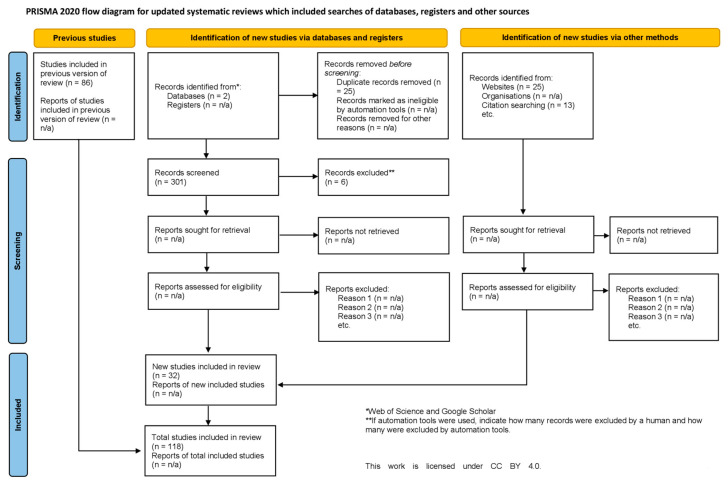
Adapted PRISMA flow diagram summarizing total articles found and total articles included in final analysis of perceptions toward wolves. To view the license for this work please visit https://creativecommons.org/licenses/by/4.0/ (accessed on 17 April 2025).

**Figure 2 animals-15-01196-f002:**
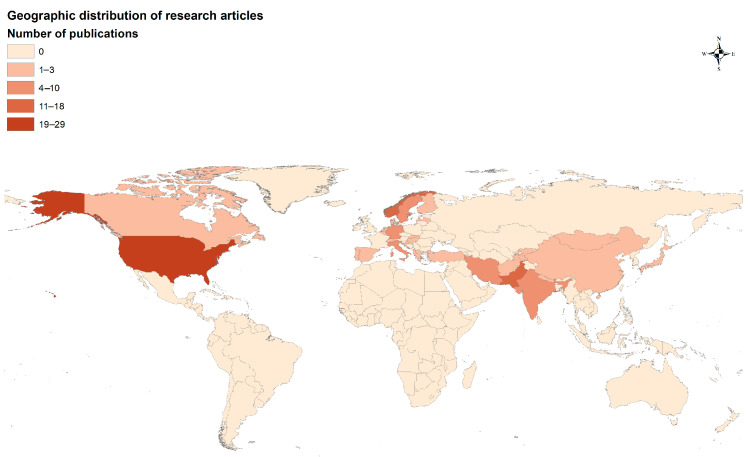
Geographic distribution of published research on perception toward grey wolves (1980–2023).

**Figure 3 animals-15-01196-f003:**
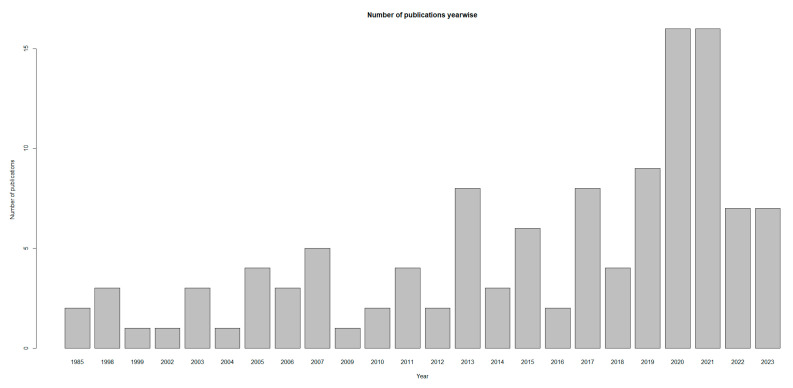
Temporal scale of published articles on perception toward grey wolves between 1980 and 2023.

**Figure 4 animals-15-01196-f004:**
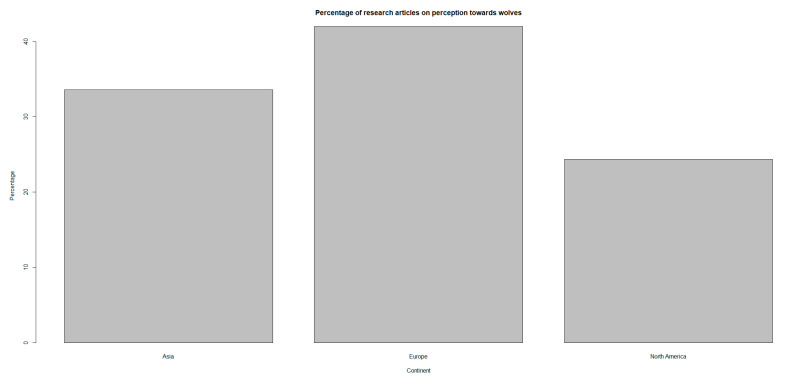
Percentage of published articles on perception toward grey wolves across continents between 1980 and 2023.

**Table 1 animals-15-01196-t001:** Primary and secondary variables extracted from research articles on perceptions toward grey wolves across their distribution range.

Variable Type	Description
Broad categories of attitudes/perceptions toward wolves (Primary)	Positive Negative Neutral
Respondent group (Primary)	Hunters, livestock farmers, livestock herders, pastoralists, students, schoolteachers, and local residents
Locality (Primary)	Rural/Urban
Religious groups (Secondary)	Hinduism/Buddhism/Islam/Christianity/Shinto
Cultural region (Secondary)	Western European, Slavic Russian, Islamic, Eastern Chinese, Indian
Gender (Primary)	Male/Female
Age group (Primary)	Range in years
Literacy (Primary)	
Damage from wolves (Primary)	Yes/No
Significant predictors of perception (Primary)	Socio-demographic, economic, and cultural variables (age, gender, education, place of residence, income, occupation, economic losses, cultural beliefs)
Year of publication (Primary)	1980–2023
Study site (Primary)	
Country (Primary)	
Continent (Primary)	Europe/North America/Asia
Economic status of country according to World Bank classification (Secondary)	Low income/lower-middle income/upper-middle income/high income
Methods used (Primary)	Questionnaire/online/mail- and postal-based, etc.
Article type (Primary)	
Journal and author name (Primary)	First author and journal name

**Table 2 animals-15-01196-t002:** Results of ordinal logistic regression regarding the effect of predictor variables on perception toward wolves.

Variable	Value	SE	t-Value	*p*-Value
Community (Hunter)	1.974	2.244	0.879	0.379
Community (Livestock farmer)	0.395	2.061	0.191	0.848
Community (Livestock herder)	−7.234	0.012	−0.059	0.952
Community (Local resident)	2.495	1.688	1.477	0.139
Locality (Rural area)	1.448	0.894	1.619	0.105
Locality (Urban area)	1.854	1.521	1.219	0.222
Religion (Christianity)	−44.962	0.953	−0.472	0.637
Religion (Hindu)	−36.307	0.003	−0.001	0.001
Religion (Muslim)	6.138	0.0847	0.033	0.973
Damage wolf attacks	−0.001	0.845	−0.001	0.999
Cultural group (Indian)	11.116	0.0184	0.060	0.952
Cultural group (Slavic Russian)	1.638	1.389	1.178	0.238
Economic classification (Low Income)	−71.477	0.070	−0.001	0.001
Economic classification (Lower Middle Income)	−53.985	0.016	−0.333	0.738
Economic classification (Upper Middle Income)	−55.167	0.025	−0.216	0.828

## Data Availability

The full database of articles considered for this review has been provided as Appendix A.

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
