# Peer review of "Perception Toward Wolves Are Driven by Economic Status and Religion Across Their Distribution Range"

_animals, 2025, doi:10.3390/ani15091196_

Round 1
Reviewer 1 Report
Comments and Suggestions for Authors
Dear Authors,
I wish to compliment with you for your very good work. The manuscript is a very well done review on a wide interest topic. The manuscript have investigated and highlight some critical and key aspects affecting human tolerance and perception towards wolf also facing new interesting and stimulating aspects.
It was a pleasure for me to read this excellent paper.
Just only few suggestions:
- Please carefully checked the scientific names of species that must be in italic
- The scientific literature cited in the Introduction should be improved by citing some historical works facing the evolution of wolf population in specific areas throught the years and the socio-econimic, environmental and cultural changes that have accompained this evolution and human attitude towards this carnivore (see for example Granlund 2018-Co-existence between Humans and wolves: A new challenge for the Old World. In D. M. Woods (Ed.), Proceedings of the 28th vertebrate pest conference (pp. 69–73). University of California; Coppola et al. 2024 - Historical research of wolf population in Pisan hills between XVII and XXI centuries: evolution of tolerance in hunters, citizens, and naturalists in a conservation perspective; Gosling et al. 2019 -Recent arrivals or established tenants? History of wolf presence influences attitudes toward the carnivore. Wildlife Society Bulletin, 43(4), 639–650. https://doi.org/10.1002/wsb.1027).
Author Response
Reviewer 1
Please carefully checked the scientific names of species that must be in italic
Response: We made changed the scientific name of all species quoted in the text in italics. Please refer to Lines 66-69, 93-94, 97, 194 in the revised MS.
The scientific literature cited in the Introduction should be improved by citing some historical works facing the evolution of wolf population in specific areas throught the years and the socio-econimic, environmental and cultural changes that have accompained this evolution and human attitude towards this carnivore
Response: We have included the citations on historical work related to the evolution of the wolf population in the specific areas as suggested by the reviewer in the revised MS. Please refer to Lines 72-82,100-119 in the revised MS. Please refer to Lines 71-77, 78-87 in the revised MS.
(see for example Granlund 2018-Co-existence between Humans and wolves: A new challenge for the Old World. In D. M. Woods (Ed.), Proceedings of the 28th vertebrate pest conference (pp. 69–73). University of California; Coppola et al. 2024 - Historical research of wolf population in Pisan hills between XVII and XXI centuries: evolution of tolerance in hunters, citizens, and naturalists in a conservation perspective; Gosling et al. 2019 -Recent arrivals or established tenants? History of wolf presence influences attitudes toward the carnivore. Wildlife Society Bulletin, 43(4), 639–650. https://doi.org/10.1002/wsb.1027).
Response: We have included these citations in the revised MS as suggested by the reviewer. They are 22,23, and 25 in the reference list. Please refer to Lines 108-110, 116-118 in the revised MS.
Reviewer 2 Report
Comments and Suggestions for Authors
The article offers a good systematic review on the perception of grey wolves across different socio-cultural and economical contexts. The study is well-structured and contributes significantly to the field of research. However, I suggest the following (minor) improvements:
- Clarification of the role of religion: the study suggests that Hinduism is associated with a negative perception of wolves, which contrasts with the general reverence for wildlife in Hindu culture. It would be beneficial to provide further discussion or additional supporting data to clarify this point. Are there any specific historical or socio-economic reasons that differentiate wolves from other revered species?
- Incorporation of qualitative insights: adding qualitative perspectives (e.g., cultural narratives or interviews) could provide a more nuanced understanding.
- Role of media: the discussion on media influence is minimal; maybe expanding this perspective could strengthen the analysis.
- Conceptual clarity: the distinction between "perception" and "attitude" should be better defined.
- Language refinement: some sentences are too complex; a minor revision for clarity and readability is recommended.
In the attached file you find some indications related to where it's best to insert/modify with respect to the above list of suggestions.

The English in this paper is generally clear and academically appropriate. But, some sentences are too complex and could be streamlined for better readability. Minor grammatical and stylistic refinements would improve clarity and fluency. A professional language edit is recommended to enhance coherence and readability. Particularly, but more generally, to the lines underlined in the pdf version attached (with point 5 mentioned).
Author Response
Reviewer 2
Clarification of the role of religion: the study suggests that Hinduism is associated with a negative perception of wolves, which contrasts with the general reverence for wildlife in Hindu culture. It would be beneficial to provide further discussion or additional supporting data to clarify this point. Are there any specific historical or socio-economic reasons that differentiate wolves from other revered species?
Response: We have provided information on the perception of wolves in Hinduism and South Asian countries. Please refer to Lines 409-427 in the revised MS.
Incorporation of qualitative insights: adding qualitative perspectives (e.g., cultural narratives or interviews) could provide a more nuanced understanding.
Response: We agree with the reviewer that adding qualitative perspectives could provide a more nuanced understand of wolves. However, our focus of this study was to quantify the dominant perception and hence we included only studies using quantitative methods. Please refer to Lines 213-216, 225-226, 229-231 in the revised MS. We acknowledge the limitation of this study in Lines 556-560 in the Conclusion part of the MS.
Role of media: the discussion on media influence is minimal; maybe expanding this perspective could strengthen the analysis.
Response: We understand the role of media influence on social perspectives and opinion of people in general. We do mention the role of media and the studies documenting their impact on people’s perception towards wolves. Please refer to Lines 126-132 in the revised MS. However not all studies on human-wolf interactions have considered the role of media in the study region nor their influence on people’s opinion. This information was not available for majority of the studies. Hence, we recognize the importance and the influence of the media but we do not include this as a variable or test their effects on perception towards wolves.
Conceptual clarity: the distinction between "perception" and "attitude" should be better defined.
Response: We have provided more information on the distinction between perception and attitude and the conceptual clarity behind these terminologies. Please refer to Lines 148-170 in the revised MS.
Language refinement: some sentences are too complex; a minor revision for clarity and readability is recommended.
Response: We have made the language refinement for the sections highlighted by the reviewer. Please refer to Lines 46-58, 97-118 and 387-402 in the revised MS.
Reviewer 3 Report
Comments and Suggestions for Authors
Review of Naha et al.’s Perception toward wolves are driven by economic status and religion across their distribution range, manuscript id animals-3517168.
Naha et al. report the results of a review of literature on attitudes and perceptions of wolves since 1985. Given the importance of attitudes toward and perception of wolves for conservation, this manuscript could make a valuable contribution to the literature and to preservation of the species. Below are some suggestions that I believe would improve the quality of the manuscript.
Just because a region predominantly prefers a specific religion or has a given socioeconomic status, it does not imply that the respondents to a given study in that region have that specific religious affiliation or the typical socioeconomic status. For example, respondents to surveys might be more highly educated than those who do not respond and socioeconomic status might therefore be higher in the respondents than those who did not respond. Thus, you need to be careful in your language describing the results and discussion. That is, you cannot state that religion predicts perceptions toward wolves, but rather must state that the predominant religion in a region predicts perceptions toward wolves.
Line 373 "We demonstrated that religion had a significant effect..." You did not systematically manipulate religion, so you cannot make such a claim. You can state that the predominate religious affiliation predicts perceptions toward wolves.
You have several null results. What is the statistical power (via simulation for the regression) of your study? Do the results indicate a lack of a relationship or the lack of statistical power for detecting a relation? Your sample size seems low for reasonable statistical power for a model with six predictors.
Lines 391, 416, 475: Especially if statistical power is low, you should not treat the failure to find a difference as evidence that there is no difference.
Author Response
Reviewer 3
Just because a region predominantly prefers a specific religion or has a given socioeconomic status, it does not imply that the respondents to a given study in that region have that specific religious affiliation or the typical socioeconomic status. For example, respondents to surveys might be more highly educated than those who do not respond and socioeconomic status might therefore be higher in the respondents than those who did not respond. Thus, you need to be careful in your language describing the results and discussion. That is, you cannot state that religion predicts perceptions toward wolves, but rather must state that the predominant religion in a region predicts perceptions toward wolves.
Response: We agree with the reviewer that the predominant religion and socioeconomic status of a region or group of respondents might not completely reflect their overall influence on the perception towards wolves for the whole country. However, we do believe that religious affiliation and economic prosperity of the country has considerable influence on wellbeing of the inhabitants and how they perceive wildlife.
We have made changes and have used the word “predominant” religion in a region as one of the predictors for perception towards wolves. Please refer to Lines 353-362, 364-366, 374-378, 387-392, 404-429, 528-529 in the revised MS.
Line 373 "We demonstrated that religion had a significant effect..." You did not systematically manipulate religion, so you cannot make such a claim. You can state that the predominate religious affiliation predicts perceptions toward wolves.
Response: We agree with the suggestions of the reviewer. We have rephrased the argument and the influence of religious affiliation on perception towards wolves in the revised MS. Please refer to Lines 404-412 in the revised MS.
You have several null results. What is the statistical power (via simulation for the regression) of your study? Do the results indicate a lack of a relationship or the lack of statistical power for detecting a relation? Your sample size seems low for reasonable statistical power for a model with six predictors.
Response: We are limited by the number of studies and the variables available for this systematic review. The sample size of this study could have low statistical power to detect a relation between the predictors. However, considering the criteria we had for screening articles; we are confident that the results of the ordinal regression reflect the global perception towards wolves and identify the major factors influencing them. We agree that increasing the sample size and conducting more human dimension research would improve our understanding of people’s perception towards wolves.
Lines 391, 416, 475: Especially if statistical power is low, you should not treat the failure to find a difference as evidence that there is no difference.
Response: Thanks for the comment. We agree that certain predictors or variables are important in influencing perception towards wolves. Several studies have documented their impact on human attitudes and perceptions towards large carnivores including wolves. However, the overall synthesis and analyses of the entire dataset from the 118 studies could not detect a significant statistical effect. We do agree that site specific mitigation measures should focus on reducing or addressing these issues to improve tolerance and perception towards the species. However, a more holistic approach or global conservation strategy should also consider the interrelationships between the ecological, economic, and social factors. Please refer to Lines 446-454, 466-480, 530-533 in the revised MS.
Round 2
Reviewer 3 Report
Comments and Suggestions for Authors
Thank you for addressing my concerns.
Lines 331-336: Please verify that "respondents" is the correct language. In other sections (3.5, 3.6, 3.7) the unit is "studies" instead of "respondents".
Line 418: I am not sure what "this narrative intensity the negative perception..." means. Should "intensity" be "intensifies"?
Author Response
Comment 1. Lines 331-336: Please verify that "respondents" is the correct language. In other sections (3.5, 3.6, 3.7) the unit is "studies" instead of "respondents".
Response: We agree that the word should be "studies" and not respondents. We measure the overall perception of a study which comprised of interviews or interactions with a large sample size of respondents. We have made this correction and replaced respondent with studies in the MS. Please refer to Line 325-338 in the revised MS.
Comment: Line 418: I am not sure what "this narrative intensity the negative perception..." means. Should "intensity" be "intensifies"?
Response: We have made this change in the revised MS. Please refer to Line 419.